# A pH-Controlled Solid Inhibitor Based on *PAM* Hydrogel for Steel Corrosion Protection in Wide Range pH NaCl Medium

**DOI:** 10.3390/molecules28031314

**Published:** 2023-01-30

**Authors:** Qing Yang, Bing Lin, Junlei Tang, Yingying Wang, Hongpeng Zheng, Hailong Zhang, Zhen Nie, Yanna Zhang

**Affiliations:** 1School of Chemistry and Chemical Engineering & Institute for Carbon Neutrality, Southwest Petroleum University, Chengdu 610500, China; 2Tianfu Yongxing Laboratory, Chengdu 610217, China; 3Key Laboratory of Optoelectronic Chemical Materials and Devices (Ministry of Education), Jianghan University, Wuhan 430056, China; 4Research Institute of Tianfu New Energy, Chengdu 610217, China; 5Research Institute of Petroleum Exploration and Development, CNPC, Beijing 100083, China

**Keywords:** solid corrosion inhibitor, oleate imidazoline, polyacrylamide, pH-controlled release, L80 carbon steel

## Abstract

To provide carbon steel a long-term corrosion protection effect in NaCl solutions with different pH values, based on poly-acrylamide (*PAM*) and oleate imidazoline (*OIM*), a solid corrosion inhibitor with the properties of pH-controlled release was synthesized. SEM, FTIR and TGA results indicated that the *OIM* inhibitors were successfully loaded into *PAM* hydrogel with a high *OIM* encapsulation content (39.64 wt.%). The *OIM* release behavior from the hydrogel structure has two stages, quick release and sustained release. The pH of solutions could affect the initial release kinetics of *OIM* inhibitors and the diffusion path in the hydrogel structure. Weight loss measurement of L80 steel in different pH solutions with *OIM@PAM* proved the inhibitor responsive release mechanism and anticorrosion performance. The inhibition efficiency of *OIM@PAM* can maintain over 80% after long-term immersion in a harsh corrosive environment (pH 3), which is much higher than the inhibition efficiency of *OIM@PAM* in a moderate corrosive solution.

## 1. Introduction

Corrosion leads to material degradation in various environments, which is due to the inter-chemical and electrochemical actions between metal substrates and environments [1]. Organic inhibitors have been wildly used to solve the corrosion issue due to their high inhibition efficiency, wild application range, good solubility and relatively low toxicity [2,3,4]. The inhibition mechanism of organic inhibitors has been explained by the formation of an adsorption film. Organic inhibitors adsorbed on metal surfaces through the delocalized electrical charge on the heteroatoms such as N, S, and O [1,5]. Many scholars have focused on the effect of inhibitor functional group type, quantity, and molecular structure on inhibition efficiency [5,6,7]. However, there are several drawbacks to the direct use of inhibitors in an aqueous corrosion environment. To ensure the corrosion protection effect of organic inhibitors, the excessive use of inhibitors will lead to a waste of resources, and the inhomogeneous distribution of inhibitors in corrosive environments will limit the long-lasting effect and anticorrosion effect of inhibitors. In addition, adding inhibitors is a high-selectivity corrosion protection method, which means the inhibitors are usually suitable for a certain material and corrosion environment. 

Recently, scholars became interested in encapsulating corrosion inhibitors into containers to extend the inhibitor protection time and enhance anticorrosion effectiveness as the drawbacks of direct using inhibitors. Many researchers try to encapsulate organic inhibitors into MOFs [8], hollow particles [9], core–shell nanofiber [10], gel materials [11], and so on. A critical application of encapsulated inhibitors is to provide self-healing properties to organic coating [8,9,10,11]. However, the low inhibitor concentration and complex construction of self-healing coating limited the industrial applications. Another important application of encapsulated corrosion inhibitors is solid inhibitors. The inhibitor release behavior from encapsulated container is adjusted according to the change of environment by modifying the inhibitor container. Wang et al. [12] reported calcium alginate gel capsules loaded with inhibitors. The synthesized capsules could release inhibitors during the sinking process, and effectively prevent the tube in oil well from corrosion. Dong and his coworkers [13,14,15] reported a series of solid intelligent inhibitors. The container could release inhibitors faster in an acidic environment in comparison with a neutral environment, and provide better anticorrosion performance. 

Hydrogel material is a hydrophilic 3D natural or synthetic polymer network structural gel, which could swell in water. Hydrogel has been used in drug delivery systems due to the controllable release rate [16]. Researchers have tried to use hydrogel in corrosion protection systems. Wen et al. [17] reported an solid hybrid hydrogel coating for steel corrosion protection. The weight percentage of loaded inhibitor is about 10%, and the releasing property of inhibitors rises as the external environment pH value decrease. Dong et al. [13] tried to use hydrogel to synthesize solid inhibitors, which benefits long-term corrosion inhibition due to the sustained inhibitor supply. Hydrogel is an ideal container material to synthesize a solid inhibitor, and still needs further investigation on increasing the inhibitor content, controlling inhibitor release behavior, and enhancing the mechanical property to make the application of a solid inhibitor based on the hydrogel. 

Owing to the advantages of excellent corrosion protection performance, low toxicity, stability and low economy cost, imidazoline and its derivatives are widely utilized in the industrial fields. The high inhibition efficiency of oleate imidazoline (*OIM*) derivatives is associated with good adsorption characteristics and the ability to form a hydrophobic film on metal surfaces [11,18]. Imidazoline and its derivatives have excellent inhibition effects in acid [19], neutral [20] and alkaline [21] mediums. In our previous work [11], *OIM* was introduced into gel coating to enhance the anticorrosion performance of coating in 3.5 wt.% NaCl. In this study, the anticorrosion performance is mainly dependent on the physical shielding effect of the coating, but the combination of gel material and corrosion inhibitor to enhance the anticorrosion performance is feasible and effective. Inspired by this work, gel material could be an ideal carrier for encapsulating material to protect carbon steel. The controllable release behavior of inhibitors could enhance the environment applicability and protection time. Therefore, the drawbacks of directly using inhibitors in an aqueous corrosion environment would be significantly improved. 

This work successfully synthesized a pH-controlled release solid inhibitor based on *PAM* hydrogel and *OIM*. Scanning electron microscope (SEM), Fourier-transform infrared spectroscopy (FTIR), thermal gravimetric analysis (TGA), and mechanical tests were employed to investigate the structure and characteristics of *OIM@PAM*. The *OIM* release behavior of *OIM@PAM* in various pH value aqueous environments was studied by the UV–visible spectrophotometer, and the release mechanism is discussed in-depth. The outstanding anticorrosion performance of *OIM@PAM* in different pH corrosion mediums was carried out by weight loss measurement and SEM observation, and the solid inhibitor might be applicable for the corrosion protection of facilities and pipelines in oil production.

## 2. Results and Discussion

### 2.1. Characteristics of OIM@PAM 

#### 2.1.1. Surface Morphology of *OIM@PAM*

Figure 1 shows the surface and interior morphologies of synthesized *PAM* hydrogel and *OIM@PAM*. It can be seen that *OIM* loaded into *PAM* hydrogel causes different morphology. For the *PAM* hydrogel (Figure 1a,b), the surface and interior morphology are relatively smooth and flat without pore canals. This result is due to the surface tension of water and the *PAM* interface, which lead to the tight bonding between *PAM* molecules during the evaporating progress [22], while the surface and interior morphology of *OIM@PAM* includes numerous pore canals, which is the characteristic structure of hydrogel containers loaded with corrosion inhibitors after vacuum drying [17]. The pores area distribution in the surface and interface of *OIM@PAM* is counted through the software of “Image–J”, and the results are displayed in Appendix A. The pore canals existing in *OIM@PAM* can provide space for *OIM* accommodation and release.

#### 2.1.2. FTIR Analysis

Figure 2 illustrates the FTIR spectra of *OIM@PAM*, and the pure *OIM* and *PAM* were also tested as control samples. The high-intensity broad absorption peak located in 3440 cm^−1^ is assigned to the antisymmetric stretching vibration of the –NH– group that existed in *OIM*, *PAM* and *OIM@PAM* [23,24]. The sharpened peak at 1640 cm^−1^ contributed to the −C=C−, −C=N− and −C=O− double bonds [12,24], which can be found in *OIM*, *PAM* and *OIM@PAM*. The stronger peak at 1640 cm^−1^ of *OIM@PAM* than *PAM* indicates the load of *OIM* corrosion inhibitor. There are several characteristic peaks only appearing in *OIM* and *OIM@PAM* FTIR spectra. The weak peaks at 2923 cm^−1^ and 2861 cm^−1^ correspond to the symmetric and antisymmetric stretching vibrations of the −CH_2_− group, respectively, which only consisted of *OIM* and *OIM@PAM* [25]. The interferential peak at 2359 cm^−1^ is associated with the antisymmetric stretching vibration of CO_2_ due to infrared spectrometer optical path imbalance [26], and the peak located in 1290 cm^−1^ is attributed to the stretching vibration of the tertiary amine group in *OIM*’s imidazole ring [12,23]. The above peaks indicate that *OIM* was successfully loaded in *PAM* hydrogel. In addition, there is no new peak in *OIM@PAM* in comparison with the FTIR spectra of *OIM* or *PAM*. The *OIM* is doped into the network structure of *PAM* without a chemical reaction, which result is consistent with the interior morphology of *PAM* hydrogel.

#### 2.1.3. Thermostability and Inhibitor Loading Content of *OIM@PAM*

Figure 3 compares the thermal gravimetric analysis (TGA) results of *OIM*, *PAM* and *OIM@PAM*. The investigated *OIM* and *PAM* hydrogel showed two main degradation stages. The first stage is related to the evaporation of water molecules and other volatile impurities through dehydration [27]. For *OIM*, the weight loss of this stage (40–190 °C) is 8.56%, and for *PAM* hydrogel, the weight loss of the first stage (40–190 °C) is 6.48%. These results reveal that the water content in *PAM* is lower than that of *OIM*. More importantly, the *OIM* has better thermal stability in comparison with *PAM*. The second stage is related to decomposition [27,28]. *OIM* started to decompose at 190 °C and achieved the maximum decomposition rate at 375 °C. The mass change of *OIM* stopped at 500 °C, the weight loss was 86.03%, and 3.60% substance remained. The decomposition of *PAM* started at about 190 °C and reached the highest decomposition rate at about 400 °C. The weight loss of *PAM* was 66.30% mainly due to the intramolecular and intermolecular imidization reactions on the amide group of *PAM*. These imidization reactions occurred when the temperature reached 190 °C, and released NH_3_, H_2_O and CO_2_ [29].

For the TGA curve of *OIM@PAM*, the first weight loss stage was at 40–190 °C. The weight loss caused by water volatilization was about 6.93%. As the temperature increased to 250 °C, the TGA curve of *OIM@PAM* was parallel to that of *PAM*. This result is due to the high thermal stability of *OIM* at this temperature, and the weight loss of synthesized composition is mainly caused by *PAM*. As the temperature further increased, the *OIM@PAM* rapidly decomposed. *OIM@PAM* had the highest decomposition rate at about 375 °C, which was consistent with *OIM*. The capacity of *OIM* in *OIM@PAM* could be calculated by the decomposition ratio of TGA curves using the following equation (1) [30]:*D_OIM_* × *C_OIM_* + *D_PAM_* × *C_PAM_* = *D_OIM@PAM_*(1)
where *D_X_* is the decomposition ratio of the compound, and *C_X_* is the content of *OIM* or *PAM* in *OIM@PAM*. Therefore, the content of anticorrosion inhibitor *OIM* loaded in the *OIM@PAM* is 39.64%, which is quite a high value in comparison with other researchers [13,31]. The TGA results reveal the excellent thermal stability of synthesized *OIM@PAM*, which could be used in the corrosion environment of temperature lower than 190 °C, and the high *OIM* load content could ensure the anticorrosion performance of this solid inhibitor, which will be discussed in the following section.

#### 2.1.4. Mechanical Properties of *OIM@PAM*

Appropriate mechanical property is the premise to ensure the practical application of *OIM@PAM* in a corrosion environment, especially for the flowing harsh corrosion environment. The stress–strain curves of *PAM* hydrogel and synthesized *OIM@PAM* are shown in Figure 4. The average tensile strength (*σ_b_*) values of *PAM* and *OIM@PAM* are 17.69 MPa and 13.21 MPa, respectively. Compared with *PAM*, the hybrid of *OIM* into *PAM* leads to the decrease of *OIM@PAM* strength and a remarkable increase of elongation at break. From the TGA, the *OIM* load content in *OIM@PAM* is 39.64%, and the *PAM* content is almost 60.36%. The hybrid of *OIM* into *PAM* leads to the monomer concentration of *PAM* decreasing and weakening the length of the *PAM* polymer chains. *PAM* with a shorter chain length leads to the decrease of the physical entanglement strength. Therefore, a lower tensile strength of *OIM@PAM* is obtained compared with *PAM* [32]. When the *OIM@PAM* sample was stretched, reversible non–covalent interactions between the *PAM* network and *OIM*, such as π–π stacking and hydrogen bond, etc., can break to effectively dissipate energy and prevent crack propagation, thus increasing its elongation at break [33].

### 2.2. Inhibitor Releasing Characteristics and Mechanism of OIM@PAM

#### 2.2.1. Release Behavior of *OIM@PAM* in Different pH Environment

Figure 5 shows the release behavior of *OIM@PAM* in wide range pH solutions. From Figure 5a, the concentration of *OIM* in the test solutions declined as the immersion time increased. In the first 24 h, the inhibitor concentration decreased from over 350 mg/L to about 100 mg/L. As the releasing time further increased, the inhibitor *OIM* concentration maintained at several dozen mg/L. The two release stages might be caused by the different releasing mechanisms. *OIM* is a highly effective corrosion inhibitor in various corrosive environments. In our previous study [11], the inhibition efficiency of 10 mg/L *OIM* in a simulated 3.5 wt.% NaCl corrosion environment could reach up to 92%. The inhibitor release behavior of *OIM@PAM* in different pH solutions had no definite difference. The released *OIM* concentration in pH 3 solution was the highest, which could reach 60 mg/L after 168 h of immersion. As the immersion solution pH increased to 7, the released *OIM* concentration after 168 h decreased to about 20 mg/L. As the solution pH value further increased, the *OIM* concentration increased to 30 mg/L. Figure 5b shows the cumulative release ratio of inhibitors from *OIM@PAM* in different pH solutions. In the first 24 h, the cumulative release ratio of the inhibitor reached about 20% to 30% for each condition. As the release time increased, the apparent release ratio could be observed. After the 168 h release test, the cumulative release ratio of the tested sample in pH 3 solution reached the highest value at 81.09%. As the test solution pH value increased, the release ratio decreased to 47.40% in a neutral environment, and then slightly increased in an alkaline solution. The release behavior of *OIM* from *OIM@PAM* is directly affected by the pH value of solutions.

#### 2.2.2. Release Mechanism of *OIM@PAM*

From Figure 5, two release stages of *OIM@PAM* could be observed [34]. When *OIM@PAM* was immersed in solutions, *OIM* could be released quickly through the short diffusion path in the first 2 h, which led to the high concentration of *OIM* in the first 2 h. The *OIM* concentration gradually decreased over 2 to 24 h in the releasing process, which was related to the enlargement of the transport path of *OIM* from the interior of *OIM@PAM* to the test solution. As the immersion time increased, the decrease of the *OIM* amount in *OIM@PAM* and the increase of the transport distance led to the released *OIM* concentration continuing to decline from 24 h to 168 h. The two release stages of the cumulative release curve were fitted by the Korsmeyer–Peppas equation [17,35,36] and the Parabolic equation [37,38], respectively.
(2)$$\mathrm{Stage}\;I\;(0\;\mathrm{to}\;24\;h):\;\mathrm{Korsmeyer}–\mathrm{Peppas}:\;\frac{M_{t}}{M_{\infty }}=kt^{n}$$
(3)$$\mathrm{Stage}\;\mathrm{II}\;(24\;\mathrm{to}\;168\;h):\;\mathrm{Parabolic}:\;\frac{{(M}_{t}/M_{\infty })}{t}=kt^{-0.5}+a$$
where *M_t_* and *M_∞_* are the cumulative release ratio of *OIM* at time t and infinite time, respectively. *k* is the release behavior kinetic constant, which is associated with the *OIM* delivery system. For the Korsmeyer–Peppas model in the first stage, *n* is an important exponent, which could determine the release mechanism of *OIM* from *OIM@PAM* [17]. If *n* ≤ 0.45, the release mechanism follows Fick diffusion, and the inhibitor release is controlled by inhibitor concentration gradient [39,40]. If 0.45 < *n* < 0.89, the release mechanism is dominated by Anomalous transport or non-Fick transport [17,41]. If *n* ≥ 0.89, the release behavior is followed Case II transport [39,41], which means the release rate is only controlled by the matrix relaxation [39]. For the second stage, a is a constant. The parabolic model of the second stage indicates a sustainable release range [38]. All fitted results are presented in Figure 6, and the fitted parameters are presented in Table 1.

Figure 7 shows the schematic of the two *OIM* releasing stages. For the first stage, the *OIM* release behavior is in good agreement with the Korsmeyer–Peppas equation, and the fitted R^2^ is above 0.99. The n values for this stage in different pH solutions are in the range of 0.45~0.89, which indicates the inhibitor release mechanism is in accord with the anomalous transport [17,41]. The inhibitor release behavior is controlled by both diffusion and matrix relaxation [39]. The released *OIM* due to the diffusion mechanism follows Fick’s law presented in Equation (4):(4)$$\frac{\partial c}{\partial t}=D\frac{\partial^{2}c}{\partial x^{2}}$$

The matrix relaxation of *PAM* gel is due to the absorption of water into the gel 3D network, and the swelling of the *PAM* network is caused by water invasion, resulting in the release of inhibitors. The *OIM* could dissolve in water. Once the *OIM* in *OIM@PAM* contacts with corrosion mediums, the dissolution of *OIM* becomes the initial dynamic force of the inhibitor release. The process of water invasion into gel also follows Fick’s second law [42,43], which means the axial water transfer is according to the concentration-dependent diffusivities in Equation (5):(5)$$\frac{\partial C_{w}}{\partial t}=\frac{\partial }{\partial z}(D_{w}\frac{\partial C_{W}}{\partial z})$$
where *C_w_* is the water concentration in hydrogel, *z* is the water transfer distance, *t* presents the time and *D_w_* is the diffusion coefficient of water in the *PAM* gel at time *t*. Since it is assumed that the diffusion coefficient depends on the solvent concentration (water in this study), the Fujita model of free volume is used to model solvent ingress kinetics [42,44] in Equation (6):(6)$$D_{w}=D_{w,eq}exp({-\beta }_{w}\left(1-\frac{C_{w}}{C_{w,eq}}\right))$$
where, *D_w,eq_* is the diffusion coefficient of water in the fully swollen *PAM*, *β_w_* is a structural parameter related to the *PAM* swelling rate, which will be further disused in 3.2.3, and *C_w,eq_* is the water concentration in the fully swollen *PAM*. In addition, this model is only concerned with the initial absorption of water, and the water concentration change in *PAM* at *t* = 0 and *x* = 0 is zero. The absorbed water into *PAM* gel could replace the inhibitor. Therefore, the inhibitor content released into solutions is equal to the volume fraction of the inhibitor in the *PAM* gel in Equation (7):(7)$$C_{inh}=\frac{V_{inh}}{V_{hydro-Gel}}C_{w}$$

The total released amount of *OIM* consists of Fick diffusion and hydrogel swelling, as shown in Equation (8):*Q_t_* = *Q_f_* + *Q_s_*(8)
where *Q_t_* is the total release amount of inhibitor, and *Q_f_* and *Q_s_* are the release amount of inhibitor followed Fick and non-Fick diffusion, respectively. For this stage, the release behavior of *OIM@PAM* shows little difference in various pH immersion solutions. On the one hand, the pH values of solutions have no influence on the release behavior of Fick diffusion. That is, the *Q_f_* of *OIM@PAM* release behavior has no difference in various immersion solutions. On the other hand, the different dissolution behavior of *PAM* in solutions with various pH values causes the difference of release behavior, and the dissolution behavior of *PAM* will be discussed later. In brief, in the first stage, the stable status of *OIM@PAM* leads to little difference in inhibitor release behavior at various pH values. 

For the second stage, the inhibitor release behavior turns into the Parabolic model. As the releasing time increased, the water penetrated into the *PAM* hydrogel and led to the swelling of the hydrogel. The major effect of water penetrant on the gel entanglement network is the inducement of viscoelastic stress [45]. During this process, water enhances the mobility of gel chains by converting the glassy matrix into a swollen material, and there are two moving fronts for this process:

(1) a sharp interface between unpenetrated gel and swollen gel (U–S interface), which propagates inwards into the gel.

(2) a gel–water interface (G–W interface), which moves outwards and progressively increases the gel layer thickness. 

Several researchers used the water volume fraction Φ(xt) in the gel layer to describe the penetration process and the moving behavior of the two fronts [45,46,47,48]:(9)$$\frac{\partial \Phi }{\partial t}=-\frac{\partial J}{\partial x}=-\frac{\partial }{\partial x}\left(-D\frac{\partial \Phi }{\partial x}+\vartheta_{sw}\Phi \right)=\frac{\partial y}{\partial x}(D\frac{\partial \Phi }{\partial x}\left(1-\Phi \right))$$
where *D* is the water diffusivity in the gel material and *ϑsw* is the swelling velocity of the gel. *x* is the distance between the U–S interface and G–W interface.

Assuming the gel material would not dissolve during the immersion process means the volume expansion is only caused by water absorption. Therefore, the absorbed water and gel are incompressible and the mixture has no volume change. That is to say:(10)$$\vartheta_{sw}\left(x,t\right)=D\frac{\partial \Phi }{\partial x}$$

For this stage, the *OIM* release has three steps. Firstly, at the fronts between unpenetrated gel and swollen gel (U–S interface), the swelling of dry gel needs an initiate threshold concentration of water [49]. Therefore, the initial release process of the inhibitor at the U–S interface is controlled by the initial swelling of the gel, and the release kinetics of the process were described in Equtions (6) and (7). Secondly, the diffusion of the inhibitor in the swollen gel is influenced by the viscoelasticity of the hydrogel structure [50]. Finally, at the gel–water (G–W) interface, the gel 3D net structure is filled with water. The inhibitor release behavior could be regarded as an equilibrium state, which is only followed by Fick’s law.

#### 2.2.3. Swelling Behavior and Micromorphology of *OIM@PAM*

The macro morphology and volume change of *OIM@PAM* before and after the 168 h releasing test are presented in Appendix A and Figure 8, respectively. Before the release test, the length of the cube shape *OIM@PAM* is 12.5 mm and the surface color is yellow, which is mainly due to the color of *OIM*. After the 168 h test, the shape of *OIM@PAM* is still a cube, which indicates the solid inhibitor could maintain the mechanical strength during the immersion time. The surface color of *OIM@PAM* faded, especially for that immersed in the pH 3 solution. In combination with the *OIM* releasing curves in Figure 5, the fade of *OIM@PAM* is caused by *OIM* releasing. The side length of the *OIM@PAM* cube after immersion grew longer significantly in comparison with the pristine cube. These results indicate that the swelling rate of *OIM@PAM* is related to the pH value of the corrosive solution. According to Figure 8, the volume change rate of *OIM* in the pH 3 solution is about 85 mm^3^/h, which is the highest one of all the test conditions, and the change rate decreases as the immersion solution pH increases to 7, and then increase as the pH further increase. *PAM* can react with H^+^ or OH^−^ in solution [51,52,53]. In the acid medium, the AM group in *PAM* has hydrolysis and imidization reactions [51]. As the medium pH decreases, the imidization reaction would be the major reaction. In the alkaline solution, *PAM* would hydrolyze into acrylic acid and ammonium salt [51,52]. Therefore, the volume change of *PAM* during immersion is composed of three factors: (1) the swelling of hydro–gel during immersion, (2) the dissolution of *PAM* material in an aqueous environment and (3) the release behavior of inhibitors. It follows from the above that the volume change of gel material is larger than the diffusion flux of water in Equation (12). This situation could enhance the inhibitor release at the interface between unpenetrated gel and swollen gel, and the initial release step of the inhibitor from *PAM* is strongly affected by the solution pH value. 

The internal micro images of the freezing-drying *OIM@PAM* after a 168 h release test in different pH were observed via SEM at 20 kV, and the results are presented in Figure 9. For all conditions, the internal morphology showed a palisading arrangement due to the evaporation of water during the freezing-drying process. The fence-like channel could be the passage of inhibitor release. The distribution of channels of a solid inhibitor in pH 3 and 11 is dense and orderly, and the distance between two channels is about 100 μm. For the inhibitor immersed in pH 5 and 9, the distance between two channels slightly increased. The distribution of channels in a *PAM* inhibitor immersed in pH 7 is much looser than the other conditions. This result is consistent with the *OIM* release results in Figure 5. As we discussed above, the diffusion behavior of the inhibitor in swollen gel is controlled by the viscoelasticity of the gel structure and Fick’s law. Once the swollen gel material forms the ordered channel, the inhibitor release rate will be enhanced, and the distribution of inhibitor pathways could also affect the release behavior. 

### 2.3. Corrosion Protection Effect of OIM@PAM in Various pH NaCl Solutions 

#### 2.3.1. Weight Loss Measurements

Figure 10a shows the corrosion behavior of L80 steel in the different pH solutions of 3.5 wt.% NaCl without *OIM@PAM*, and the relevant data are listed in Appendix A. The corrosion rates of L80 steel in test solutions without a solid inhibitor have high values, especially in the acid solutions. As the immersion time increases, the corrosion rate of L80 steel in the test solutions slightly decreases. After the 168 h corrosion test, the corrosion rate of L80 steel remains above 0.2 mm/y, which is classified as severe corrosion for steel in pH 3 and 5 and high corrosion for steel in neutral and alkaline solutions, according to the NACE–RP0775 standards. The unprotected steel in the test solution underwent serious deterioration resulting from electrochemical corrosion. The corrosion mechanism of steel is changed by the pH value of the corrosive medium. In an acid solution, a localized electrochemical reduction–oxidation reaction is the main corrosion mechanism [54]. For steel immersed in neutral and alkaline NaCl solutions, the corrosion rate increases with the pH values augmented [55]. The corrosiveness of 3.5 wt.% NaCl solution at different pH for L80 steel long-term soaking follows: pH 3 > pH 5 > pH 11 > pH 9 > pH 7. Therefore, the corrosion protection requirements of L80 steel in various pH value test solution are different, which requires the release of a solid inhibitor that could adapt to the environment and provide an appropriate corrosion protection effect.

Figure 10b shows the corrosion rate of L80 steel in test solutions with *OIM@PAM*, and the inhibition efficiency of *OIM@PAM* is calculated. The corrosion rate of L80 steel immersed in inhibited test solutions is dramatically decreased in contrast with solutions without solid inhibitors. The released *OIM*, as an imidazoline derivative, could absorb on the steel surface and form a hydrophobic film to suppress the corrosion reactions on steel surface [1,11]. N atom in the imidazole ring could provide electrons to the steel surface, enhancing the adsorption effect of *OIM* on the steel surface. The oleate carbon chain tail in *OIM* would help to increase the hydrophobicity of the absorption film [56]. The good inhibition performance of *OIM* is the foundation of the solid inhibitor. For the steel immersed in acid solution (pH 3 and 5), the corrosion rate decreased obviously in comparison with the other conditions. After 168 h of immersion, the corrosion rate of L80 steel in the pH 3 and 5 test solutions was 0.0432 mm/y and 0.0511 mm/y, respectively. For the steel immersed in neutral and alkaline solutions, the corrosion rate was also reduced to below 0.076 mm/y according to Chinese standards SY/T 5329. The different inhibition effect of *OIM@PAM* is associated with the corrosivity of the test solution and the release characteristics of *OIM@PAM*. The inhibition efficiency (IE%) calculated by the weight loss results are also presented in Figure 10b and Appendix A. The highest IE% value for each condition was obtained at 24 h due to the quick release of inhibitors from *OIM@PAM* in the first release stage (Figure 5). As the immersion time increased, the IE% slightly decreased, especially for the pH 7 solution. This result was due to the release difference of *OIM@PAM* in various pH solutions. According to the above discussion, the solid inhibitor of *OIM@PAM* could provide an appropriate corrosion protection effect for steel in different corrosive mediums, especially for an acid environment.

#### 2.3.2. Surface Observation of L80 Steel after Immersion Test

The Raman spectrum is used to study the adsorption of *OIM* on a L80 steel surface after the immersion test, and the results are shown in Figure 11. The pure *OIM* molecule has three characteristic peaks. The peaks located at 1662 cm^−1^, 1440 cm^−1^ and 1302 cm^−1^ are attributed to vibrations of the C=N double bond, C−N bond and −CH_2_− group, respectively [11]. For the L80 steel immersed in a solution without *OIM@PAM*, there are characteristic peaks represented by γ−FeOOH (at 250 cm^−1^ and 389 cm^−1^), α−Fe_2_O_3_ (at 298 cm^−1^) and Fe_3_O_4_ (at 675 cm^−1^) [57]. For the L80 steel immersed in the pH 7 solution with *OIM@PAM*, the characteristic peaks caused by corrosion products and *OIM* could both be found. This result indicates the *OIM* released from *OIM@PAM* could be absorbed on a steel surface and provide excellent long-term protection performance for steel. 

The SEM micrographs of L80 steel after the 168 h immersion test in uninhibited and inhibited pH 3, 7 and 11 solutions are presented in Figure 12. According to Figure 12a, the surface of L80 steel corroded in the pH 3 test solution looks similar to the lunar surface. The rough surface contains several pits with diameters about 30 μm. The surface of steel immersed in pH neutral and alkaline are presented in Figure 12b,c. The corrosion morphology in these conditions shows similar characteristics, which means the corrosion mechanism of these conditions is same. After adding *OIM@PAM*, the corrosion evidence on the steel surfaces of all conditions shows a significant reduction. The surface roughness of steel decreases in comparison with the uninhibited solution. For the steel in the pH 3 inhibited solution, there are some white dots and polishing traces on the steel surface. The surfaces of steel immersed in pH 7 and 11 inhibited solution are similar, while the polishing traces on the surface of steel in pH 11 is more obvious than that of steel in pH 7. The SEM results further confirmed the excellent inhibition effect of *OIM@PAM* in NaCl solution with a wide pH range, especially for the more corrosive environment.

## 3. Materials and Methods

### 3.1. Synthesis of OIM@PAM Solid Corrosion Inhibitor

Acrylamide (AM), N,N-methylene-bis-acrylamide (BIS), ammonium persulfate (APS), tetramethylethylenediamine (TEMED) and oleate imidazoline (*OIM*) all purchased from Shanghai Macklin Biochemical Technology Co., Ltd. (Shanghai, China) for the preparation of the solid corrosion inhibitor.

Firstly, as Figure 13 shows, 1.88 g acrylamide (AM) was dissolved into 5 mL deionized water. Secondly, 1 g oleic imidazoline (*OIM*) was dispersed to the above solution with continuous stirring until the inhibitor dissolved completely, followed by adding 0.032 g N,N-methylene-bis-acrylamide (BIS) to the mixed solution, and all of the above steps were processed under the temperature range of 0–4 °C to prevent acrylamide from polymerizing prematurely. Thirdly, 0.015 g ammonium persulfate (APS) was introduced into the above solution to convert acrylamide monomers to free radicals. The free radicals would react with unactivated monomers to begin the polymerization chain reaction, while the addition of TEMED aims to accelerate the rate of formation of free radicals from APS and, consequently, catalyze the polymerization. After that, the as-prepared mixed solution was poured into a silica gel mold (20 mm × 20 mm × 20 mm) and polymerized at 45 °C for 12 h. Finally, after drying in a vacuum oven (D2T–6050, Jinghong Experimental Equipment Co., Ltd., Shanghai, China) at 45 °C for 24 h, the solid corrosion inhibitor was successfully synthesized and recorded as *OIM@PAM*. The average weight of obtained *OIM@PAM* is 2.5583 g, and the side length is 12.5 mm.

### 3.2. Characterization Methods

The micromorphology of *OIM@PAM* was observed by scanning electron microscope (ZEISS EV0 MA15, Carl Zeiss, Dublin, CA, USA) with the accelerating voltage of 20 kV. The structure of *OIM@PAM* was measured by a fourier-transform infrared analyzer (Nicolet 6700, Thermo Scientific, Waltham, MA, USA) with a wavelength range of 500 cm^−1^–4000 cm^−1^. Thermo-gravimetric analysis (DSC823, METTLER TOLEDO, Greifensee, Switzerland) was employed to estimate the loaded amount of *OIM* at a heating rate of 10 °C/min in a temperature range of 40–600 °C in the N_2_ atmosphere. The tensile test of *OIM@PAM* mainly referred to the ASTM standard D822 at room temperature in atmosphere, which was measured by an electronic universal testing machine (ETM502C, Wance Co. Ltd., Shenzhen, China) with the crosshead speed of 5 mm/min [58]. Each sample was measured three times to guarantee the accuracy of the results. A raman spectrometer (BWS465–785S, B&W TEK, Newark, DE, USA) was used to study the *OIM* adsorption behavior at the L80 steel/corrosive solution interface. The selected laser wavelength was 785 nm. 

### 3.3. Release Behavior of OIM from OIM@PAM

A UV–Visible spectrophotometer was used to investigate the release behavior of *OIM* from *OIM@PAM*. The standard *OIM* solutions with concentrations of 50, 100, 125, 150, 175 and 200 mg/L were prepared. The UV–Vis curves of the standard *OIM* solution are shown in Figure 14a, and the relationship between absorbance and *OIM* concentrations was linearly fitted and is shown in Figure 14b. 

To investigate the *OIM* release behavior from *OIM@PAM*, *OIM@PAM* was completely immersed in 100 mL solution with various pH (3, 5, 7, 9 and 11). After a certain time (2 h, 4 h, etc.), 5 mL immersion solution was taken out and used to test the *OIM* concentration by UV–Vis, and the *OIM* release amount can be calculated according to *OIM* standard curve (Figure 14). At the same time, the rest solution was replaced by the appropriate pH solution to simulate the flow state of the corrosive medium. According to the results of the releasing test and TGA test, the cumulative release ratio of *OIM* from *OIM@PAM* can be calculated.

### 3.4. Corrosion Protection Performance of OIM@PAM

The L80 carbon steel (wt. %: 0.36%C, 0.45%Si, 1%Mn, 0.03%P, 0.004%S, 0.25%Ni, 0.38%Mo, and balance Fe) purchased from China Jiangsu Xinyou Instrument Co., Ltd. (Changzhou, China) was used as the corrosion substrate. The surfaces of L80 coupons were polished with 160# to 2000# SiC grit paper [59] and rinsed with acetone, deionized water and ethanol. The 3.5 wt.% NaCl solutions with different pH values (3, 5, 7, 9 and 11) were selected as the corrosive medium. The above-mentioned reagents were purchased from Chengdu Kelong Chemical Reagent Factory (Chengdu, China).

The corrosion protection performance of *OIM@PAM* on L80 steel in a wide range pH value environments was tested by a weight loss experiment. The original qualities of the L80 samples were recorded using an analytical balance with a precision of ±0.1 mg. Then, the L80 samples were immersed in different pH values of 3.5 wt.% NaCl solution with and without *OIM@PAM*. The above experiments were carried out at 25 °C, and the different pH value corrosive solutions were replaced every 24 h to simulate the flow state of the mediums.

Corrosion products were removed using an acid-washing solution composed of 10% HCl + 0.5% ammonioformaldehyde (C_6_H_12_N_4_) [60]. After that, each L80 sample was weighted three times through electronic balance to ensure the reliability of the tested data. The corrosion rate (*CR*) and inhibition efficiency (*IE*) of L80 are calculated according to the following equations [61]:(11)$$CR=\frac{\Delta w\times 87600}{St\rho }$$
(12)$$IE\left(\%\right)=\left(1-\frac{CR}{CR_{0}}\right)\times 100$$
where Δ*w* with the unit of gram (g) is the mass diffidence of L80 before and after the experiment. *S* with the unit of square centimeter (cm^2^) is the surface of L80. *t* with the unit of hour (h) is the experiment time of L80. *ρ* with the unit of gram per cubic centimeter (g·cm^−3^) is the density of L80. *CR* and *CR_0_* with the unit of millimeter per year (mm·y^−1^) are the corrosion rate of L80 in the medium with and without *OIM@PAM*, respectively.

## 4. Conclusions

(1) *OIM* is successfully loaded into the *PAM* gel network, and the load amount is up to 39.64%. The synthesized *OIM@PAM* has good thermal stability, which could be used in an environment below 190 °C, and *OIM@PAM* also has good mechanical properties. 

(2) The release behavior of *OIM* from *OIM@PAM* depends on the external solution pH values, and its release has two stages. The first stage is the *OIM* quick release from the *PAM*, which is followed by Fick’s law and hydro-gel swelling. The second stage is the *OIM* sustained release, when the release rate is controlled by the initiate threshold concentration of water for the un-swollen gel and the inhibitor pathway in the swollen gel. 

(3) The corrosion protection performance of the *OIM@PAM* solid inhibitor in 3.5 wt.% NaCl solutions with a wide range of pH values is checked through weight loss measurement. The corrosion rate of L80 steel in the NaCl solution can be reduced to below 0.076 mm/y, and the IE% for the *OIM@PAM* in all conditions are higher than 80%. The Raman and SEM results further confirmed the corrosion protection effect of *OIM@PAM*.

## Figures and Tables

**Figure 1 molecules-28-01314-f001:**
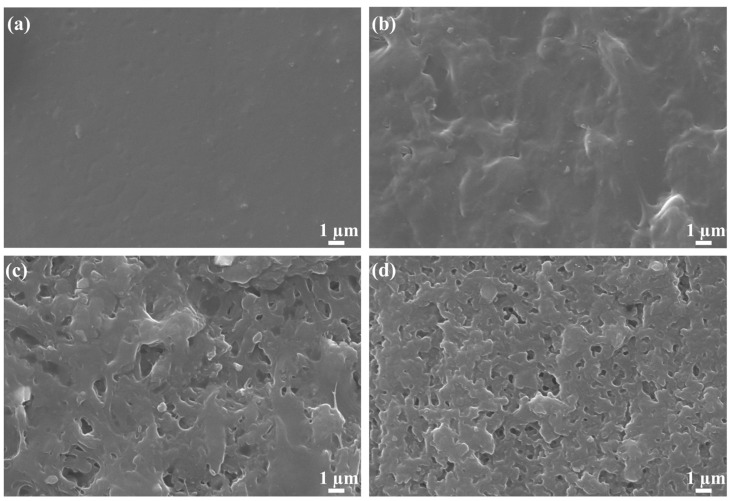
The surface (**a**) and interior (**b**) SEM micrographs of the as-prepared *PAM* after vacuum-drying; The surface (**c**) and interior (**d**) SEM micrographs of the as-prepared *OIM@PAM* after vacuum-drying.

**Figure 2 molecules-28-01314-f002:**
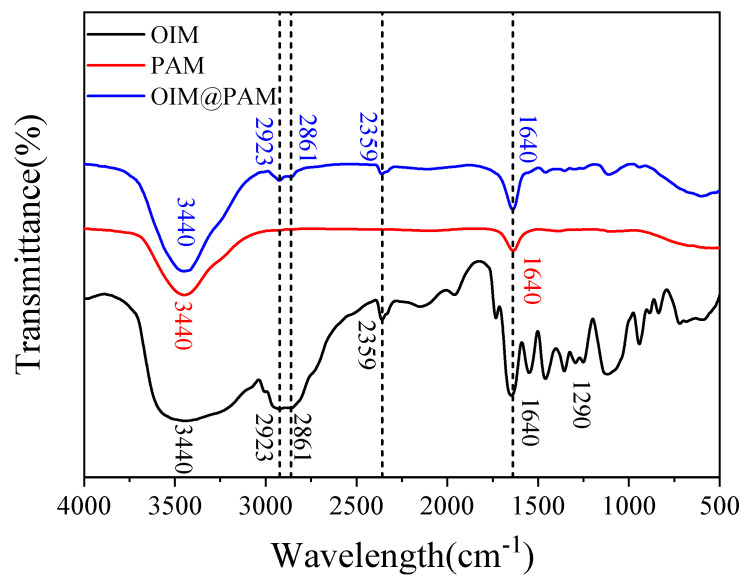
FTIR spectra of pure *OIM*, *PAM* and *OIM@PAM*.

**Figure 3 molecules-28-01314-f003:**
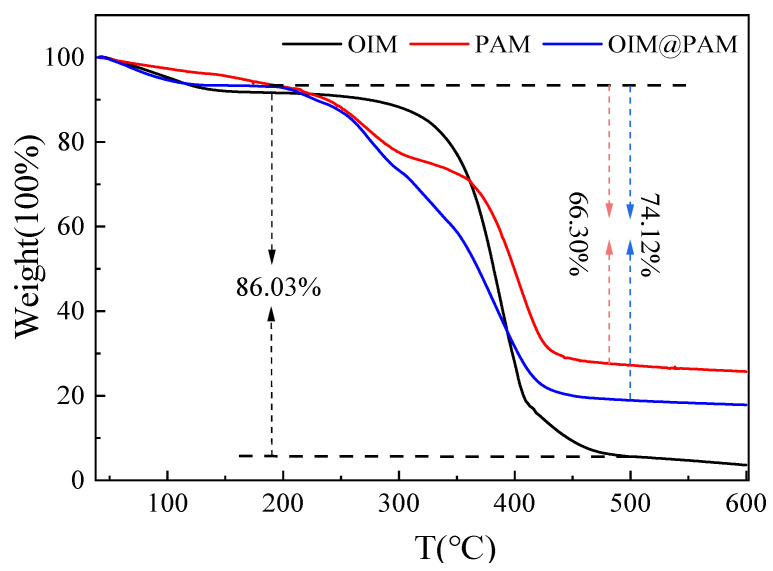
TGA curves of pure *OIM*, *PAM* and *OIM@PAM*.

**Figure 4 molecules-28-01314-f004:**
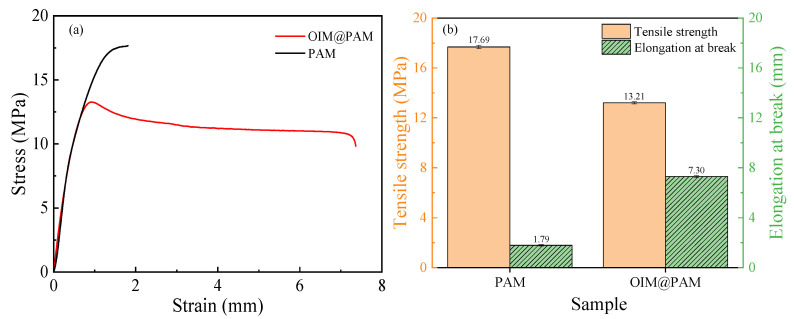
(**a**) Tensile stress–strain curve of *PAM* and *OIM@PAM* after vacuum-drying for 24 h at 45 °C; (**b**) the values of the tensile strength and the elongation at break of *OIM* and *OIM@PAM*.

**Figure 5 molecules-28-01314-f005:**
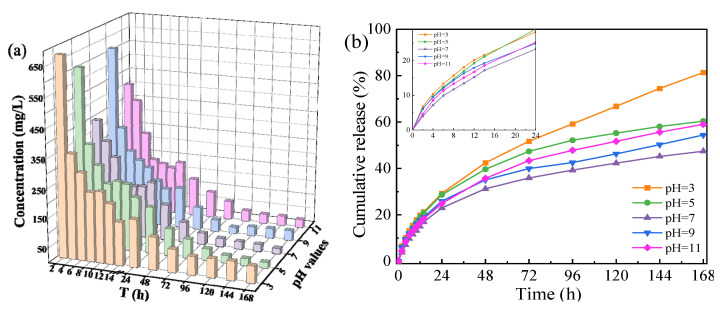
The three-times-tested average concentration (**a**) and the cumulative release (**b**) of *OIM* from *OIM@PAM* in different pH solutions at 25 °C.

**Figure 6 molecules-28-01314-f006:**
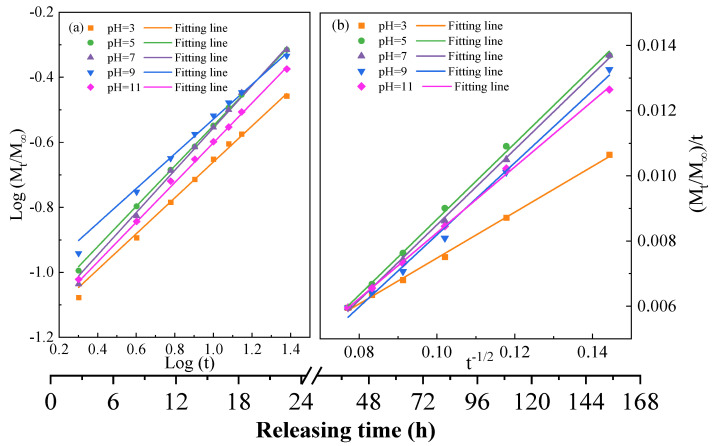
Plots of the two kinetic models: (**a**) Korsmeyer–Peppas model and (**b**) Parabolic model for the release of *OIM* from *OIM@PAM* in the two stages.

**Figure 7 molecules-28-01314-f007:**
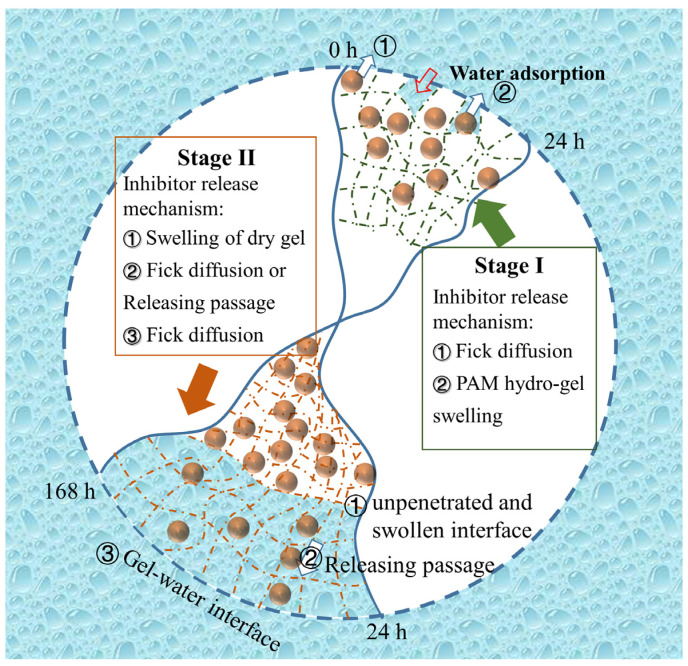
Schematic of two releasing stages of the inhibitor from *OIM@PAM*.

**Figure 8 molecules-28-01314-f008:**
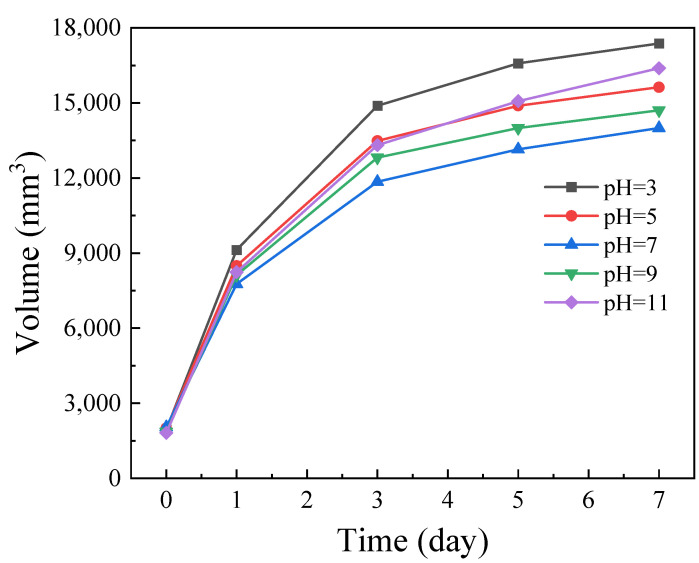
The change in volume of *OIM@PAM* immersed in various pH test solutions for 7 days.

**Figure 9 molecules-28-01314-f009:**
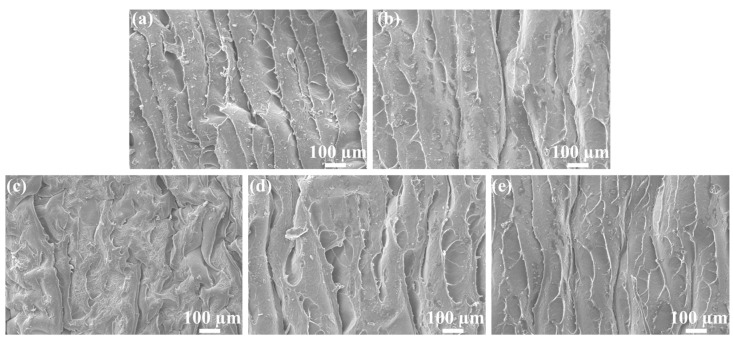
The internal SEM images of *OIM@PAM* after releasing for 168 h in different pH solutions. (**a**) pH 3, (**b**) pH 5, (**c**) pH 7, (**d**) pH 9 and (**e**) pH 11.

**Figure 10 molecules-28-01314-f010:**
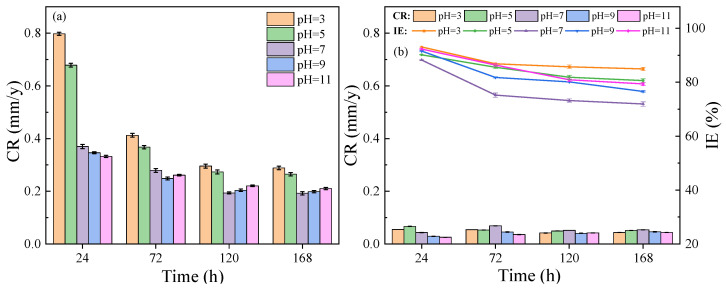
Corrosion rate and inhibition efficiency of L80 in different pH solution of 3.5 wt.% NaCl at 25 °C with and without *OIM@PAM*, (**a**) without *OIM@PAM* and (**b**) with *OIM@PAM*.

**Figure 11 molecules-28-01314-f011:**
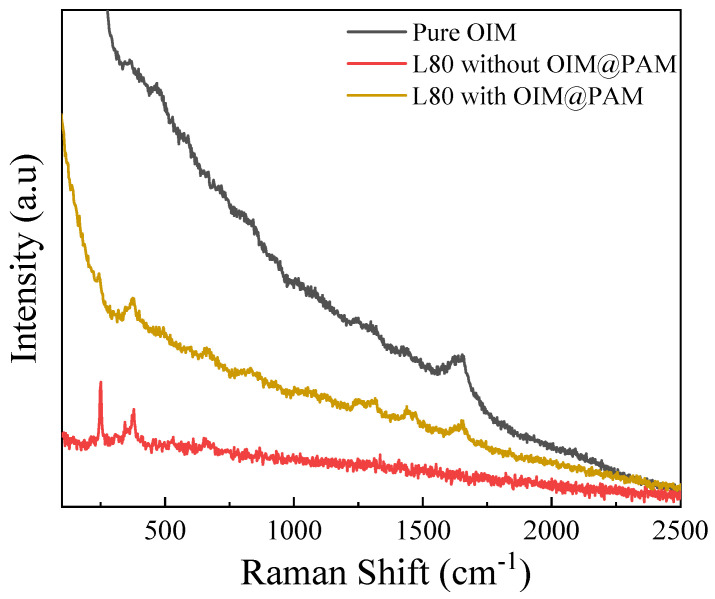
Raman spectrum of pure *OIM* and L80 immersed in 3.5 wt.%NaCl with and without *OIM@PAM* after 72 h.

**Figure 12 molecules-28-01314-f012:**
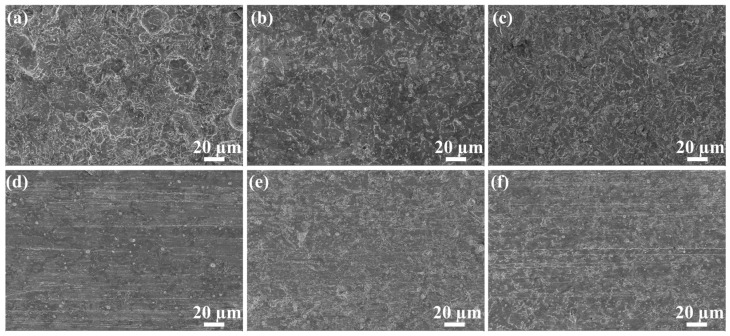
SEM micrographs of the L80 steel surface immersed in test solutions for 168 h: (**a**) pH 3 uninhibited solution, (**b**) pH 7 uninhibited solution, (**c**) pH 11 uninhibited solution, (**d**) pH 3 solution with *OIM@PAM*, (**e**) pH 7 solution with *OIM@PAM* and (**f**) pH 11 solution with *OIM@PAM*.

**Figure 13 molecules-28-01314-f013:**
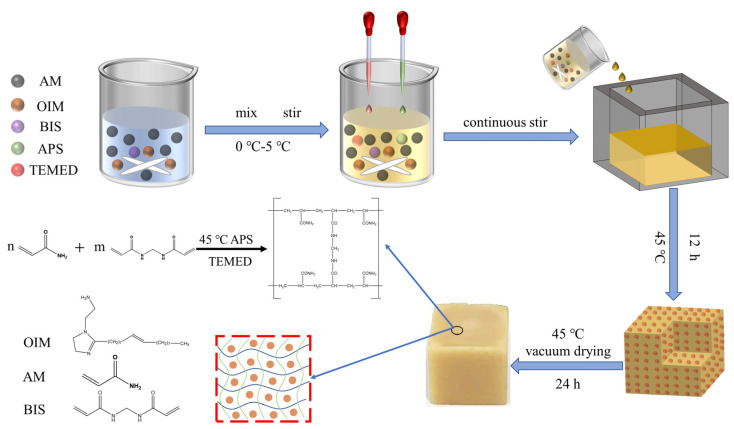
Schematic diagram of the synthesis of *OIM@PAM* solid corrosion inhibitors.

**Figure 14 molecules-28-01314-f014:**
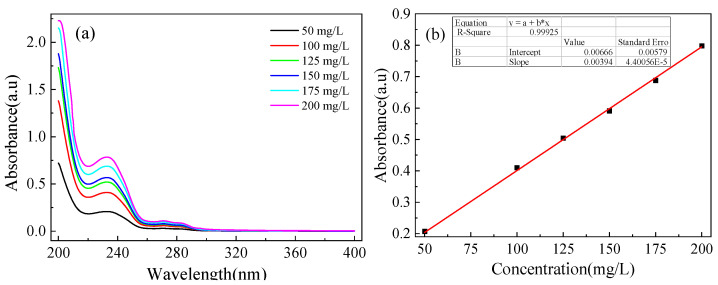
(**a**) UV–Vis spectra for different *OIM* solutions of known concentrations, and (**b**) the fitted standard curve of *OIM*.

**Table 1 molecules-28-01314-t001:** The fitting parameter and mechanisms of *OIM* from *OIM@PAM* in different pH solutions.

pH Value	Stage 1 (0~24 h)				Stage 2 (24~168 h)			
	Korsmeyer–Peppas Model				Parabolic Model			
	n	k	R^2^	Release Mechanism	k	a	R^2^	Release Mechanism
3	0.5340	0.00351	0.9952	Anomalous transport	0.07034	0.00044	0.9978	Sustainable release
5	0.7435	0.0028	0.9962	Anomalous transport	0.1304	−0.00413	0.9998	Sustainable release
7	0.6514	0.0033	0.9949	Anomalous transport	0.1156	−0.00307	0.9988	Sustainable release
9	0.5526	0.0612	0.9960	Anomalous transport	0.1106	−0.00288	0.9918	Sustainable release
11	0.6114	0.0033	0.9939	Anomalous transport	0.1010	−0.00185	0.9984	Sustainable release

## Data Availability

The synthesized *OIM@PAM* samples and all raw/processed data necessary for reproducing the results in this study can be accessed upon reasonable request.

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
