# Peer review of "A pH-Controlled Solid Inhibitor Based on PAM Hydrogel for Steel Corrosion Protection in Wide Range pH NaCl Medium"

_molecules, 2023, doi:10.3390/molecules28031314_

Round 1

Reviewer 1 Report

This manuscript has systematically studied the intelligent corrosion inhibitor based on oleate imidazoline and poly-acrylamide hydro-gel. The synthesized intelligent inhibitor has a high drug loading capacity, which could reach up to 40%. The authors have described the inhibitor molecular release process form PAM gel based on the immersion test and surface observation. The release mechanism is important for the release behavior of small molecular drugs form gel material. The synthesized intelligent inhibitor exhibited a good anti-corrosion performance in a wide pH range, especially in harsh corrosion environment. Therefore, I recommend the manuscript could be accepted after a minor revision.

1.      There are still some grammatical errors, error reference and misspell in the manuscript. For example: 1)      The 45th reference. 2)      The first time of abbreviation in line 86 needs full name. 2.      The authors should use the same terminology throughout the manuscript. Such as, Line 23-25, the “inhibitor” represents the OIM@PAM intelligent inhibitor. But the most of “inhibitor” in the manuscript represents OIM inhibitor. 3.      Line 160, the authors replace the corrosive solution every 24 h, the experimental details should be described in detail. 4.      OIM inhibitor and L80 steel in NaCl solution was tested in this work, whether there is a specific industrial background or application requirements?

5.      In Fig. 3, the surface and interface of PAM loaded with OIM shown “numerous pore canals”. Could statistics method apply to the SEM images to quantization the area ratio of pore canals?

Author Response

Point 1. There are still some grammatical errors, error reference and misspell in the manuscript. For example:1)     The 45th reference. 2)     The first time of abbreviation in line 86 needs full name.  

Response 1. Thanks to your comments. We have overviewed the manuscript to correct the mistakes in the manuscript, and the details are as follow:

1) The duplicate serial number of the 45th reference is amended due to our careless.

2)  and the abbreviated terms of “SEM”, “FT-IR”, “TGA” are rewritten as Scanning electron microscope (SEM), Fourier Transform infrared spectroscopy (FT-IR), Thermal Gravimetric Analysis.

3)  In addition, all of the grammatical and spelling errors of the manuscript is carefully checked and corrected by an English native professor, the detailed corrections are in Appendix located in end of this cover letter. 

Point 2. The authors should use the same terminology throughout the manuscript. Such as, Line 23-25, the “inhibitor” represents the OIM@PAM intelligent inhibitor. But the most of “inhibitor” in the manuscript represents OIM inhibitor.

Response 2. Thanks so much for your suggestion, there are four incorrect expressions about the terms of “inhibitor” in manuscript. The corrections are as follows. 

1) Original: Line 24. which is much higher than the inhibition efficiency of inhibitor in moderate corrosive solution.

Correction: which is much higher than the inhibition efficiency of OIM@PAM in a moderate corrosive solution. 

2) Original: line 405. The side length of inhibitor cube after immersion has grown longer significantly in compare with the pristine cube.

Correction: The side length of OIM@PAM cube after immersion has grown longer significantly in comparison with the pristine cube. 

3) Original: line 464. Fig 12 (b) shows the corrosion rate of L80 steel in test solutions with OIM@PAM, and inhibition efficiency of inhibitor is calculated.

Correction: Fig 12 (b) shows the corrosion rate of L80 steel in test solutions with OIM@PAM, and the inhibition efficiency of OIM@PAM is calculated. 

4) Original: line 465-467. The corrosion rate of L80 steel immersed in inhibited test solutions is dramatically decreased in contrast with in solution without inhibitor.

Correction: The corrosion rate of L80 steel immersed in inhibited test solutions is dramatically decreased in contrast with solutions without intelligent inhibitors. 

Point 3. Line 160, the authors replace the corrosive solution every 24 h, the experimental details should be described in detail.  

Response 3.  Thanks so much for your suggestion. We have accurately described the corrosive solutions and given reasons for regular replacement based on your comments in line 165-167. The changes are as follows:“The above experiments were carried out at 25 ℃, and the different pH value corrosive solutions were replaced every 24 h to simulate the flow state of mediums”. 

Point 4. OIM inhibitor and L80 steel in NaCl solution was tested in this work, whether there is a specific industrial background or application requirements? Response 4. Thanks so much for your comments. As far as we know, L80 carbon steel is widely used due to its advantages of cost-effective and suitable mechanical strength in oil and gas fields. Furthermore, we analyze corrosive ions and its content in formation water. Therefore, 3.5 wt% NaCl solutions are selected to simulate the formation water environment in oil fields. For OIM inhibitors, it is widely used in CO2 corrosion protection of carbon steel in oil fields. However, the drawbacks of direct use inhibitors, including short-term protection effect, often cause trouble to operators. Therefore, the characteristics of OIM@PAM intelligent inhibitor make it possible to use in the production process of oil and gas fields. 

Point5. In Fig. 3, the surface and interface of PAM loaded with OIM shown “numerous pore canals”. Could statistics method apply to the SEM images to quantization the area ratio of pore canals?

Response 5. Thanks so much for your comments. We use the software of “Image J” to calculate the pores area distribution of OIM@PAM intelligent inhibitor. The area of pores on the OIM@PAM surface and interface is concentrated between 0.02-0.2 µm2 and 0.02-0.1 µm2, respectively. The total area of pores on the surface and interface is 14.98 µm2 and 18.96 µm2, respectively. Based on your suggestion, we added this discuss on supplementary materials.

Reviewer 2 Report

The manuscript examines the inhibitor release from PAM hydrogel. The obtained results show differences in inhibitor release with pH but the problem is that the change in inhibitor release is not the same as the change in medium corrosivity which is something expected from an intelligent inhibitor system. Although the highest amount of the inhibitor is released for pH3, where the highest corrosion rate is observed, for pH5, with the second highest corrosion rate the amount of the released inhibitor is similar to that of  the less corrosive pH9 (figure 7a). For that reason the title intelligent inhibitor is too strong.

One of the characteristics of intelligent inhibitors is that they respond to pH change, did you examine what is happening when the pH of solution changes in time?

The second problem with the manuscript is that there is no comparison between corrosion inhibition with PAM hydrogel system and the corrosion inhibition when the inhibitor is added directly to the solution (as done in reference 11.).  Such experiment would prove benefits of using hydrogel system.

The language of the manuscript needs significant improvement, as there are many errors and ambiguous sentences.

Author Response

Thanks so much for your comments and suggestions to our manuscript entitled “A pH-controlled intelligent inhibitor based on PAM hydrogel for steel corrosion protection in wide range pH NaCl medium”. These comments and suggestions are highly insightful and helpful for us to improve the quality of our work, and the suggestions are helpful to guide our researchers. We have revised the manuscript carefully according to your comments and suggestions, and the responses to your comments are as follows:

Point 1. The manuscript examines the inhibitor release from PAM hydrogel. The obtained results show differences in inhibitor release with pH but the problem is that the change in inhibitor release is not the same as the change in medium corrosivity which is something expected from an intelligent inhibitor system. Although the highest amount of the inhibitor is released for pH3, where the highest corrosion rate is observed, for pH5, with the second highest corrosion rate the amount of the released inhibitor is similar to that of the less corrosive pH9 (figure 7a). For that reason, the title intelligent inhibitor is too strong.

Response 1. Thanks to your comments. We also notice that the similar release amount of OIM@PAM in pH=5 and pH=9. Three times parallel release experiments of OIM@PAM are carried out. the same results of parallel experiments were gotten. To our best knowledge, there is no precise definition about the “intelligent inhibitor”. The manuscript is entitled “Intelligent Inhibitor” mainly based on the following consideration. In the first place, the pH-responsive release properties of OIM@PAM are generally consistent with the solution corrosiveness. The release of OIM originates from the interaction between the PAM hydrogel and the mediums. Furthermore, the corrosion rates of L80 carbon steel in wide range pH value solutions can be controlled below 0.076 mm/y, which is safe for the production process of oil and gas fields according to Chinese standard SY/T5329. Therefore, we guess that the title of “intelligent inhibitor is reasonable. 

Point 2. One of the characteristics of intelligent inhibitors is that they respond to pH change, did you examine what is happening when the pH of solution changes in time?

Response 2. Thanks to your comments. The change of the release characteristics of the PAM system in different pH solutions is due to PAM gel hydrolysis [1–3]. Therefore, it has certain response properties to the solution change. This manuscript focuses on the OIM@PAM intelligent inhibitor released in different single pH solutions, and its corrosion protection performance. The release process of OIM@PAM is still studying when the pH of solution changes in time, And will be reflected in next work.  

Point 3. The second problem with the manuscript is that there is no comparison between corrosion inhibition with PAM hydrogel system and the corrosion inhibition when the inhibitor is added directly to the solution (as done in reference 11.).  Such experiment would prove benefits of using hydrogel system. Response 3. Thanks to your comments. the experiments of studying corrosion protection performance of directly adding OIM to NaCl solution have been adequately studied through the method of weight loss measurement, electrochemical test and theoretical calculation in our other work, and published in journal of “Corrosion Science” [4,5]. The work [4] mainly focused on using gel coating to encapsulate OIM to enhance the anti-corrosion performance of coating in 3.5 wt.% NaCl. But the anti-corrosion performance is mainly dependent on the physical shielding effect of the coating. Based on previous experimental studies, this manuscript is designed to avoid the drawbacks of gel coating and directly use of inhibitor. 

Point 4. The language of the manuscript needs significant improvement, as there are many errors and ambiguous sentences. 

Response 4. Thanks so much for your comments. All of the grammatical and spelling errors of the manuscript is carefully checked and corrected by an English native professor. The detailed corrections are in Appendix located in end of this cover letter, which are highlighted in orange color.

Round 2

Reviewer 2 Report

The manuscript is improved from the previous version but still some issues are not fully addressed:

Regarding the previous comment on studies in solution authors have responded: „the experiments of studying corrosion protection performance of directly adding OIM to NaCl solution have been adequately studied through the method of weight loss measurement, electrochemical test and theoretical calculation in our other work, and published in journal of “Corrosion Science” [4,5].“ However, cited reference 4 is not from Corrosion Science journal, nor any of them is published by the authors or describe above mentioned results. So please provide correct references and reflect on the comparison between inhibition in solution and by „intelligent inhibitor“ in the manuscript.

If authors insist on keeping intelligent inhibitor in the title, then they should explain in text why this system can be considered as such.

Please check one more time the text as there as still some ambiguous sentences.

Author Response

Thank you very much for your comments, your insightful comments are so important for us to improve the quality of the manuscript. We have carefully revised the manuscript according to your comments, and the responses to your comments are as follows:

Point 1. Regarding the previous comment on studies in solution authors have responded: „the experiments of studying corrosion protection performance of directly adding OIM to NaCl solution have been adequately studied through the method of weight loss measurement, electrochemical test and theoretical calculation in our other work, and published in journal of “Corrosion Science” [4,5].“ However, cited reference 4 is not from Corrosion Science journal, nor any of them is published by the authors or describe above mentioned results. So please provide correct references and reflect on the comparison between inhibition in solution and by “intelligent inhibitor” in the manuscript.

Response 1. Thanks a lot for your comments. The protective effect of adding OIM for carbon steel in NaCl solution has been adequately studied and published in the journal of “Corrosion Science” [1,2]. We are sure that the cited refernce1, entitled “Self-healing performance of ethyl-cellulose based supramolecular gel coating highly loaded with different carbon chain length imidazoline inhibitors in NaCl corrosion medium”, is from the journal of “Corrosion Science”. It can be found at https://www.sciencedirect.com/science/article/abs/pii/S0010938X22000026.

The work [1] primarily studies the synergistic mechanism and the synergistic effect of oleic imidazoline (OIM) and L-cysteine (CYS) for carbon steel protection in NaCl solution. The results of the weight loss and the electrochemistry experiment show that the inhibition performance of the mixture corrosion inhibitors is better than that of the individual inhibitor. When OIM with the concentration of 50 mg/L was tested by mass loss measurement, electrochemical impedance spectroscopy and potentiodynamic polarization measurements, the inhibition efficiency is 61.01%, 75.48% and 86.91%, respectively. The different values of corrosion inhibition efficiency are caused by the different test methods. The results of quantum chemical calculation indicate that the capability of donating and accepting electrons of OIM is better than that of CYS.

The work [2] mainly focus on using supramolecular gel coating to encapsulate OIM with different carbon chain (C7, C11 and C17). The detailed experimental results of adding the liquid corrosion inhibitor OIM are as follows (the first paragraph of the fifth page of the article). The electrochemical measurements result of 20# steel in 3.5 wt. % NaCl solution without or with 10×10-4 mol/L inhibitor are shown in Figure 4[2]. The results, including EIS and PPCs, all confirm that the anti-corrosion performance of imidazoline inhibitor increases as the carbon chain length of inhibitor increases. The inhibition efficiency reaches 91% in the test solution with C17. The EDS results indicate that the corrosion protection performance originates from the adsorption of the N atom.

The corrosion inhibition performance of the OIM@PAM solid corrosion inhibitor only comes from the release of its component of OIM. The function of PAM hydrogel system is to control the release progress of OIM inhibitor through the interaction between itself and the environment, thus obtaining a long-term (0 h-168 h) corrosion protection effect. The long-term corrosion protection effect of only adding liquid OIM could not be obtained due to the fluidity of the solutions. Based on the previous work [1,2], the corrosion inhibition performance and corrosion inhibition mechanism of OIM on carbon steel in NaCl solution have been well studied. This manuscript mainly focuses on the synthesis of solid corrosion inhibitors, the pH-controlled release process and mechanism of solid corrosion inhibitors, and the long-term corrosion inhibition efficiency of OIM@PAM in wide range of pH values.

Point 2. If authors insist on keeping intelligent inhibitor in the title, then they should explain in text why this system can be considered as such.

Response 2. Thank you so much for your comments. In conjunction with your comment, we have carefully checked the experimental results of our manuscript again and again. some inappropriate really exists about the inhibitor of OIM@PAM entitled as intelligent inhibitor. We revise the description of “intelligent inhibitors” in the manuscript to “solid inhibitor” according to the feature of OIM@PAM.

Point 3. Please check one more time the text as there as still some ambiguous sentences.

Response 3. Thanks so much for your meticulous work. We carefully checked the manuscript again, and the use of verbs in wrong tenses and some ambiguous sentences have been revised by an English native professor. The detailed corrections highlighted in blue color are as follows.

The detailed responses are in file.
